# The Roles of Clinical Psychologists in Burns Care: A Case Study Highlighting Benefits of Multidisciplinary Care

**Anna V. Cartwright \*** and **Elizabeth Pounds-Cornish**

Burns Unit, Stoke Mandeville Hospital, Buckinghamshire Healthcare NHS Trust, Aylesbury HP21 8AL, UK
\* Correspondence: bht.burnspsychosocialteam@nhs.net; Tel.: +44-1296-315040

**Abstract:** The British National Burn Care Standards highlight the importance of routine psychosocial screening to optimise psychological well-being following burn injury. Routine screening enables clinicians to identify those who may benefit from further psychological intervention. In this case, we outline how active follow-up from routine psychosocial screening and early intervention supports psychological recovery from a burn injury and how multidisciplinary care can be incorporated into cognitive therapy for post-traumatic stress disorder. This case also illustrates how psychologists are well positioned within physical healthcare to notice themes arising in patient care and use this to inform service development, for example, through staff training.

**Keywords:** psychosocial care; psychosocial screening; acute stress disorder; post-traumatic stress disorder; trauma-focused cognitive therapy; multidisciplinary work; training

## 1. Introduction

According to the British National Burn Care Standards [1], psychosocial screening of inpatients who have sustained a burn injury, and are admitted for more than 24 h, should be completed as soon as is clinically appropriate as part of routine clinical care. Similarly, the American Burn Association (ABA) require burn centres to include brief psychological screening and intervention and recommends that all patients are screened for acute stress disorder and Post Traumatic Stress Disorder (PTSD), with follow-up provided for those with positive screens [2]. However, due to issues such as a lack of mental health providers and lack of funding, not all institutions are able to meet these standards [3].

Screening enables timely identification of psychological needs, thus supporting early intervention and prevention of longer-term distress. There is a wide range of screening and assessment tools for psychological distress that are used with burn survivors [4]. However, there is no single burns-specific standardised psychosocial screening measure.

Acute stress reactions are common following burn injury and are considered normal reactions to stress [5]. For example, a prospective study of patients following major burn injury reported a point prevalence of 23.6% of acute stress disorder during hospitalisation within the first month of a traumatic event [6]. Acute stress disorder is characterised by the presence of nine or more symptoms from the following categories that persist for between 3 days and 1 month after trauma exposure and cause clinically significant distress or impairment: intrusions (such as recurrent, involuntary, and intrusive distressing memories of the traumatic event), negative mood, dissociation, avoidance (for example of memories, thoughts or feelings associated with the event, and/or external reminders), and arousal symptoms (such as sleep disturbance, irritability, hypervigilance, difficulties concentrating). If symptoms persist for more than 1 month, this may be indicative of PTSD. In a sample of patients admitted for treatment of a major burn injury, 35.1% were found to meet the criteria for PTSD one month after discharge [6]. Importantly, the presence of acute stress disorder post-injury is a risk factor for PTSD [7]. Therefore, psychosocial screening shortly following injury provides an opportunity to identify those who may require further psychological treatment after discharge.

The National Institute for Health and Care Excellence (NICE) guidelines recommend offering individual trauma-focused Cognitive Behavioural Therapy (CBT) interventions, such as Cognitive Therapy for PTSD (CT-PTSD), to adults with acute stress disorder or clinically important symptoms of PTSD [8]. There is robust evidence for the clinically important effects of trauma-focused CBT [9]. Psychologists in burns units are well-placed to offer these interventions in a timely manner following burn injury.

Psychosocial factors have also been found to be associated with burn recovery, further highlighting the importance of early psychosocial screening as part of multidisciplinary inpatient care. For example, those with a pre-existing psychiatric diagnosis or higher levels of post-burn psychological distress have been reported to have longer hospital stays and require more surgical procedures than those without psychiatric history or high levels of psychological distress [10].

The NICE 4-level stepped care model of psychological assessment and intervention [11], developed for use in cancer services, can be applied across many healthcare contexts, including burns care. This model emphasises how good psychological care is the role of all healthcare professionals and that having a psychologically informed workforce reduces the likelihood of psychological distress developing. Psychological professionals have a wide range of competencies to facilitate this model of service delivery within physical healthcare. For example, in addition to direct evidence-based clinical work, clinical psychologists are trained in offering consultation to other healthcare professionals, facilitating staff support, providing education and training, designing and conducting research and audit projects, and leadership and management [12].

The purpose of this report is to illustrate the multi-faceted role of clinical psychologists in acute healthcare by outlining a patient's journey through the burns psychology service, highlighting the benefits of multidisciplinary patient care.

## 2. Case Presentation

Daniel, a Black British male in his 50s, was admitted to the burns ward after his block of flats was set on fire. He had managed to escape, with some of his neighbours, by jumping from a window. Daniel sustained 1.75% total body surface area (TBSA) superficial partial thickness burns to his hands and face.

### 2.1. Inpatient Screen and Follow-Up

Daniel was screened by the first author (AC), a clinical psychologist who had 1 year of experience working in the burns unit. The screening took place 5 days after his injury. Unfortunately, due to the absence of the only full-time member of the burns psychology team, it was not possible for a psychologist to meet Daniel prior to this date. During the psychosocial screen, Daniel described good social support from friends, his partner, and family. He reported generally coping with stress by using humour, trying to actively change things, and smoking. Daniel described himself as an independent person, a 'helper' who did not want to burden others. He reported that during the fire, he thought he would die and described feeling very upset by reminders of what had happened, having nightmares, and experiencing frequent, distressing flashbacks.

Daniel was given psychoeducation about trauma responses to normalise his symptoms as a common reaction to an extremely frightening event. He was taught grounding strategies, and feedback was provided to the multidisciplinary team. Sharing psychoeducation resources and grounding strategies with ward staff was important to ensure that information to normalise trauma symptoms was shared, and staff were able to provide consistent advice and recommend evidence-based strategies to support Daniel in managing this acute distress. Daniel was discharged to temporary emergency accommodation after 5 nights in the hospital. During a follow-up call 26 days after his injury, Daniel described persistent, frequent flashbacks, feeling afraid to go to sleep, and avoiding reminders of the incident. He reported being offered therapy with another service but chose to continue working with

the same burns psychologist, as he did not want to explain what had happened 'all over again' and felt more trust towards the hospital service.

### 2.2. Assessment

Forty-eight days after his injury, Daniel attended a psychological assessment appointment with AC. He described vivid flashbacks of himself running through the fire. He explained how he often smelt fire and petrol, felt 'frozen in time', and did not want to go to sleep in case there was another fire. Daniel reported often listening out for noises, keeping the lights on at night, and constantly feeling on guard. He described how some of his friends and neighbours had died in the fire, reported feeling guilty that he survived when others did not, and grieving for those who had died. Daniel avoided speaking to others involved in the fire and stayed away from reminders of it.

On the Impact of Events Scale-Revised (IES-R) [13], Daniel scored 66 (above the clinical cut-off of 33, indicating the likely presence of PTSD). He scored 18 on the Patient Health Questionnaire-9 (PHQ-9) [14] measure of depression ('moderate' range) and 19 on the Generalized Anxiety Disorder Scale-7 (GAD-7) [15] measure of anxiety ('severe' range). A 12-item Responses to Intrusions Questionnaire-short form (RIQ-s) [16] was used to assess the extent to which Daniel engaged in particular responses to intrusive memories, such as suppression, rumination, and numbing; a 7-item Safety Behaviours Questionnaire (SBQ) [16], to identify extra precautions Daniel was taking; and an 8-item Memories Questionnaire (MQ) [16], to obtain further information regarding the characteristics of the trauma memory.

### 2.3. Intervention

Daniel attended 14 sessions of CT-PTSD with AC, based on the cognitive model of PTSD [17]. The first three sessions involved providing psychoeducation, developing an individualised case formulation, setting treatment goals, identifying triggers for trauma symptoms, practising grounding strategies and developing the rationale for trauma-focused therapy. Daniel described wanting to return to the gym and spa but was concerned that his skin was too fragile due to the burns. In order to determine whether Daniel was medically fit to return to these activities, an advanced nurse practitioner and Burns Speciality Registrar joined a subsequent therapy session to review Daniel's skin. It was confirmed that Daniel could return to the gym and spa, and he was given advice regarding scar management. This integration of medical and psychological care helped to provide Daniel with advice and reassurance regarding his physical recovery, which gave him confidence that he was medically fit to return to these previously enjoyed activities. This facilitated further discussion regarding Daniel's thoughts and beliefs about returning to these activities, which were further explored using cognitive and behavioural interventions.

The remaining therapy sessions involved imaginal reliving of the trauma, identification of the worst moments ('hot spots'), discussion of thoughts and beliefs associated with these key moments, and cognitive restructuring to identify alternative perspectives, which were then incorporated into the reliving, to update the trauma memory. Daniel practised stimulus discrimination and used behavioural experiments to test out predictions regarding changing his sleep routine and reducing avoidance of reminders of the incident. Alongside these trauma-focused interventions, Daniel worked towards values-based goals to increase his daily activities, including working towards a return to work, spending time with family, and socialising with friends. During the penultimate session, Daniel revisited the site of the fire.

During therapy, Daniel reflected on his experiences as a Black British man. He described the racist abuse that he had been subject to over the course of his life and the impacts this has had. He explained how prior to therapy, he had 'given up on professionals'. Daniel spoke about how showing emotions can be perceived as a sign of weakness and that others make assumptions about him as a Black male of being 'strong' and, therefore, not in need of their help. Daniel described key moments during the incident and his treatment where he

had been treated unfairly because of assumptions made about him based on his ethnicity, gender, and age. For example, he described how bystanders at the scene stepped over him on the floor, a moment that often came back to him in flashbacks. Daniel also described how healthcare staff had encouraged him not to have pain medication, which he believed would have been different if he was not a Black man.

### 2.4. Outcome

In the final session, Daniel completed a therapy blueprint to summarise the work and reflect on the progress made. He described a significant reduction in the frequency and intensity of flashbacks, having techniques to manage memories, upsetting thoughts and emotions relating to the incident, improvements in his sleep, and being more able to do the things that were important to him. He set new goals and considered how to manage future setbacks, such as the anniversary of the incident. Daniel's scores on the IES-R reduced over the course of therapy to 13 (below clinical cut-off) in the final session (Figure 1).

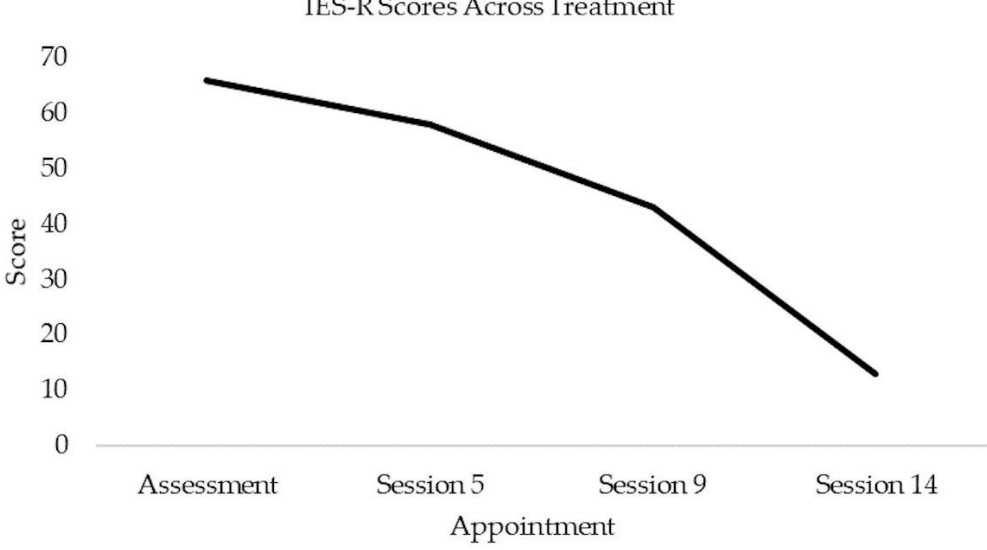

**Figure 1.** Scores on the Impact of Events Scale-Revised over the course of therapy.

At the end of therapy, Daniel scored 0 on the PHQ-9 (below clinical cut-off) and 2 on the GAD-7 (below clinical cut-off). His scores on the RIQ-s, SBQ, and MQ also reduced (Table 1).

**Table 1.** Questionnaire scores at assessment and end of treatment.

| Questionnaire | Assessment | End of Treatment |
|:---:|:---:|:---:|
| IES-R | 66 | 13 |
| PHQ-9 | 18 | 0 |
| GAD-7 | 19 | 2 |
| SBQ | 20 | 6 |
| RIQ-s | 19 | 4 |
| MQ | 9 | 1 |

IES-R = Impact of Events Scale-Revised; PHQ-9 = Patient Health Questionnaire-9; GAD-7 = Generalised Anxiety Disorder Scale-7; SBQ = Safety Behaviours Questionnaire; RIQ-s = Response to Intrusions Questionnaires-short form; MQ = Memories Questionnaire.

### 3. Discussion

This case study demonstrates the benefits of inpatient psychosocial screening in identifying those who may benefit from trauma-focused interventions following burn injury. By identifying Daniel's symptoms early and offering proactive follow-up after discharge, it was possible to offer a timely treatment and prevent ongoing symptoms and

distress. Daniel reported a preference for having trauma-focused therapy with the burns psychology service rather than another service, highlighting the importance of offering choice and consistency in support. Establishing rapport and building a trusting therapeutic relationship is a key foundation in psychological therapy.

This case highlights the importance of effective multi-disciplinary collaboration in patient care. Liaising with nursing staff on the ward was important in ensuring joined-up care for Daniel during his inpatient stay. A more timely screen would have enabled this to be put in place sooner, but it was unfortunately not possible in this case due to staffing. Working closely with medical colleagues during outpatient follow-up, who assessed Daniel's scars and provided advice that he was medically fit to return to the gym, provided reassurance that he could re-engage in previously enjoyed activities.

### 3.1. Stepped Care Model of Psychological Assessment and Support

The NICE 4-level stepped care model of psychological assessment and intervention [11] emphasises how having a psychologically informed workforce reduces the likelihood of psychological distress developing. This includes compassionate and effective communication and training clinicians to consider patients' needs holistically by integrating psychological and social factors, in addition to physical factors, into the clinical assessment. This approach enables psychological distress to be identified and appropriately addressed. To effectively implement this stepped-care approach, the model emphasises the important role of psychological professionals within acute healthcare in disseminating psychological skills to the workforce and providing supervision, support, and training.

The stepped care model has many similarities to recommendations in the British National Burn Care Standards [1]. For example, the Burns Care Standards state that all members of the MDT should receive training in psychological care appropriate to their role [18]. The aim is for all staff to be able to recognise distress signals and for relevant staff to be able to deliver the appropriate support, information, and advice to meet patients' needs. These standards also state that burns staff should have access to regular, reflective practice, as required, facilitated by members of the psychosocial care team.

Clinical psychologists are in a privileged position to discuss the details of patients' experiences of care and often facilitate training sessions with other members of the team. Importantly, themes that arise over time in conversations with patients may indicate areas for improvement in patient care. These themes may be shared with the team whilst ensuring that patient confidentiality is maintained. During therapy, Daniel spoke about the assumptions that people made about him, based on his ethnicity, gender, and age, and the impact this had on the care he received from others during and after the fire, including whilst in the hospital.

Racial and ethnic inequalities in healthcare are well documented. For example, there is consistent evidence that pain is often underestimated and undertreated in Black people relative to White people [19]; Black patients have been found to be less likely to be given pain medication and, when given, to receive lower quantities than White patients [20]. This has important implications for medical treatment, safety, and patient experience of care. Upon discussing this theme in clinical supervision, the psychology team considered how this could be incorporated into future psychosocial training sessions with the wider healthcare team to improve patient care and help reduce the risk of medical trauma. As a result, discussion and reflection regarding how we can be aware of and challenge our assumptions and biases when supporting patients in managing acute pain were incorporated into a psychosocial training session on pain management facilitated by AC and an assistant psychologist.

### 3.2. Limitations

It is important to acknowledge that this is a single case study, and although it is evident from this case that effective multidisciplinary care and consistency of psychological care had benefits for Daniel with regard to his burns treatment and recovery, the findings

cannot be generalised to other patients. Due to the single case study design, it is also not possible to determine the extent to which Daniel's recovery was a result of the intervention provided. However, given that Daniel's symptoms persisted beyond the first month post-injury without treatment, the rate at which his symptoms decreased during psychological therapy, and strong evidence for the effectiveness of CT-PTSD, it is considered unlikely that the reduction in symptoms is entirely due to spontaneous recovery.

## 4. Conclusions

There is strong and consistent evidence supporting the effectiveness of trauma-focused CBT in treating PTSD [8]. This case study highlights the importance of psychosocial screening in identifying individuals who may benefit from psychological therapy following burn injury. It highlights the value of multidisciplinary collaboration during outpatient psychological therapy when working with those who have had physical injuries to ensure that therapy goals are safe and in line with physical recovery and rehabilitation. Finally, this report outlines the varied roles of clinical psychologists within multidisciplinary teams and how anonymised themes that arise in clinical practice can be incorporated into staff training to benefit patient care.

**Author Contributions:** Conceptualisation, A.V.C. and E.P.-C.; methodology, A.V.C.; writing—original draft preparation, A.V.C.; writing—review and editing, A.V.C. and E.P.-C.; supervision, E.P.-C.; project administration, A.V.C. All authors have read and agreed to the published version of the manuscript.

**Funding:** This research received no external funding.

**Institutional Review Board Statement:** Not applicable.

**Informed Consent Statement:** Written informed consent has been obtained from the patient(s) to publish this paper. A pseudonym has been used throughout.

**Data Availability Statement:** Not applicable.

**Conflicts of Interest:** The authors declare no conflict of interest.

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
