# Peer review of "The Roles of Clinical Psychologists in Burns Care: A Case Study Highlighting Benefits of Multidisciplinary Care"

_2673-1991, doi:10.3390/ebj4010010_

Round 1

Reviewer 1 Report

Thank you for promoting the importance of psychological care among survivors or burn injuries as well as emphasizing the importance of interprofessional care and the need for concomitant psychological and phsycial healthcare. I only have a few comments that you might consider to improve this manuscript.

1. In the introduction, it might help justify the need for psychological care if you cite a source or sources that explain the impact of decreased mental health or prior psychiatric history on burn recovery. Here is a systematic review as one example that also contains many sources: O’Brien, K.H., & Lushin, V. (2019). Examining the impact of psychological factors on hospital length of stay for burn survivors: A systematic review. Journal of Burn Care and Research 40(1), 12-20. https://doi.org/10.1093/jbcr/iry040

2. In the case report, you mentioned that the patient had key moments of when he was treated unfairly based on assumptions made because of his ethnicity, gender, and age and that he had given up on professionals. I would be more clear here if the experiences of racism were from healthcare or medical professionals and what that impact may have had on his medical treatment, length of recovery, adherence to treatment, etc. Medical racism can have a significant impact on all of the above and being clear that this experience as from health providers would be important vs. a broad statement that may refer to society as a whole (treated in public, local communities, etc.). While this may not necessarily need to be cited, you may find the outcomes of this study interesting, specific to the significant discrepancies in stress levels among white burn patients for burn patients of color: O’Brien, K. (2020). Dimensions of burn survivor distress and its impact on hospital length of stay: A National Institute on Disability, Independent Living, and Rehabilitation Research Burn Model System study. Burns 46(2) 286-292 https://doi.org/10.1016/j.burns.2019.12.007

McKibben JB, Ekselius L, Girasek DC, Gould NF, Holzer C, Rosenberg M, et al. Epidemiology of burn injuries II: psychiatric and behavioural perspectives. Int Rev Psychiatry 2009;21:512 21.

Blakeney PE, Rosenberg L, Rosenberg M, Faber AW. Psychosocial care of persons with severe burns. Burns 2008;34 (4):433 40, doi:http://dx.doi.org/10.1016/j.burns.2007.08.008.

Klinge K, Chamberlain DJ, Redden M, King L. Psychological adjustments made by postburn injury patients: an integrative literature review. J Adv Nurs 2009;65:2274 92, doi:http://dx. doi.org/10.1111/j.1365-2648.2009.05138.x.

Wisely J. The impact of psychological distress on the healing of burns. Wounds UK 2013;9(3):14 7.

3. You may want to add a limitations section. While we want to promote the positive impact multidisciplinary/interprofessional care can have on patient outcomes (i.e. the change scores on the assessment tools you used), it's good practice to acknowledge this is a single subject case study, change scores can be a result of the "time-effect"/time passing or perhaps a combination of good care and the time effect. 

4. It was great you included the measurement/assessment tools. 

5. Can you explain a bit more the integration of care and emphasize its intentionality? The examples seem to be found more so in the discussion versus the intervention section - it may be more appropriate to include in the intervention and to provide a bit more depth and rationale - why having a physician attend a treatment about a patient's concern about their skin and returning to work, the gym, etc. and how a mental health provider can then use this information and collaboratively target a patient's anxiety or worry. 

6. In addition to Psychologists, you might also want to make mention of the benefit of integrating care from other mental health providers including social workers, occupational therapists, psychiatric nurse practitioners, psychiatrists, etc. 

Thanks again for writing this manuscript and raising some very important aspects of care that all members of a multidisciplinary should be aware of, if not to address themselves, to know enough to assesses for and refer to another member of the multidisciplinary team.

Reviewer 2 Report

Thank you for the invitation to review this interesting case study. I have a few suggestions that I hope will help you to make some minor improvements, but overall I think it is a useful case study to add to the body of evidence that early psychological screening, assessment and therapy is a useful addition to standard treatment, and thus recommend acceptance after some minor amendments.

Introduction

The stepped care model is the core part of the discussion, yet it does not appear in the introduction. A sentence or two need to be added to the introduction to help pave the way for the discussion.

Line 2 – It appears that the BBA standards don’t ‘offer’ a psychosocial screen, but instead specify that a screen should be completed for all patients admitted for >24h, and that this should be done as soon as is clinically appropriate. I recommend that this wording is changed to better reflect the recommended standards.

Case presentation

A new section at the beginning that described the event itself would help the reader with context around the circumstances of injury and potential stressors that are occurring in conjunction with the burn. This is important in the scenario which suggests a small burn (1.75% TBSA) which was associated with deliberate arson (‘his home was set on fire’) with intended and deliberate harm to the patient. The reader then mentally assesses whether this was a targeted attempted murder to one person, but then finds out there where other people who had lost their lives. Thus, if section 2.1 contains information about “The Event” and 2.2 is then “Inpatient Screen and Follow-up” etc the attempt by the reader to cognitively piece together the scenario and stressors involved would be resolved.

Section 2.2 Assessment reports 3 weeks after followup, and followup is 3 weeks after discharge. Instead of these timepoints, it would be more helpful to report length of stay and time since burn for more accuracy.

References

Reference years: 1997 2000 2001 2004 2006 2008 2010 2013 2016 2018 2018 2020

Of the 13 references (one is repeated), seven are more than 10 years old. It would be good to have a greater number of references to back up this case study, and perhaps some that are more recent. I also had some issued with the provided links.

·         Reference #1 the link didn’t work for me – and it appears to be here:

Microsoft Word - BCSO 2018 (FINAL).docx (britishburnassociation.org)

·         Reference #5 the link didn’t work for me – and it appears to be here:

https://www.nice.org.uk/guidance/ng116

which converts to this link when copied: Overview | Post-traumatic stress disorder | Guidance | NICE

·         Reference #6 is repeated as reference #14

·         Reference #7 – there is a 2nd edition (2004) of this book with chapter 7 having the same title – perhaps update this reference.

·         Reference #10 the link didn’t work for me – and it appears to be here after registration and login: OxCADAT ResourcesResources for cognitive therapy for PTSD, social anxiety disorder and panic disorder.PTSD Questionnaires

·         Reference #12 the link didn’t work for me – and it appears to be here:

https://www.nice.org.uk/guidance/csg4/resources/improving-supportive-and-palliative-care-for-adults-with-cancer-pdf-773375005

which converts to this link when copied:  Palliative care Adults CSG REP (nice.org.uk)

I would like to wish you well in progressing this important work.

Reviewer 3 Report

I very much enjoyed reading this paper, which reports a case study of a Black British male who sustained a burn injury in a house fire. This very well written report demonstrates how psychosocial screening identified his need for input from a clinical psychologist and clearly outlines the nature and process of that involvement, including the use of relevant measures to chart his progress.  I think this paper will make a valuable addition to this special edition. It will appeal both to clinical psychologists in this field, but also to others in burn care and research who might not be familiar with the ways in which clinical psychologists actually carry out their work within a multi disciplinary team.   

I spotted a couple of small typos (patients' rather than patient's on page 4, and the relevance of the asterisk after Daniel's name in the text was unclear), and it would be interesting to know if the same psychologist was involved throughout and the extent of their prior experience of working in burns, but otherwise I have no further comments - I would be very pleased to see it published as it stands. 
